# Consumers’ Channel Preference for Fresh Foods and Its Determinants during COVID-19—Evidence from China

**DOI:** 10.3390/healthcare10122581

**Published:** 2022-12-19

**Authors:** Xujin Pu, Jingyi Chai, Rongtao Qi

**Affiliations:** Department of Management Science and Engineering, Business School, Jiangnan University, Wuxi 214122, China

**Keywords:** channel preference, fresh foods, COVID-19, online channel

## Abstract

The public has been experiencing unprecedented challenges during the COVID-19 pandemic for the past two years. Government measures, such as improvements in offline markets and the encouragement of contactless e-commerce use, have been taken to abate the spread of infection. This study explored whether public channel preferences for fresh foods have changed and aimed to identify potential determinants. Data from 10,708 consumers were obtained by issuing questionnaires, and the binary logic measurement model was used for the empirical analysis to study the core factors that determine consumers’ choice of online and offline purchase channels for fresh food. The results show that, from the perspective of consumers’ personal behavior, consumers who do not pay attention to online evaluations and consumers who do not buy products based on their purchase experience have increased the frequency of online fresh food purchases during the epidemic. Food safety also significantly affects consumers’ choices of purchase channels. Consumers who believe that online fresh foods are safer prefer to purchase fresh food online. Among the factors affecting the performance of online fresh food, consumers concerned about food safety increased the frequency of online purchases, while consumers concerned about the reputation of the platform decreased the frequency of online purchases. These findings can help online and offline retailers better understand consumer needs and then determine their marketing strategies.

## 1. Introduction

Along with the development of e-commerce, online shopping has provided another purchase channel outside the traditional shopping environment, and the online purchasing of goods has become the mainstream consumption for consumers [1]. The COVID-19 pandemic has created new opportunities for fresh food e-commerce, which keeps some social distance with other consumers when purchasing fresh foods. Due to the outbreak of the epidemic, many American residents tried online grocery shopping for the first time in 2020. Compared with the same period during the previous year, online food orders were estimated to have increased by 700% in the first quarter of 2020 after the pandemic [2]. The total sales of the fresh food e-commerce industry in the United States have experienced a 54% increase in 2020 [3]. In Canada, before the spread of COVID-19, only 1.5% of groceries were sold online, and by the third week of March, this number had grown to more than 9.0% [4]. In fact, grocery chains have reported a surge in online orders of up to 300% [5].

Therefore, the epidemic has increased consumers’ attempts at using fresh food e-commerce platforms. According to the consumer survey of China’s COVID-19 epidemic released by market research agency Kantar, online retail channels will also become more important. During the pandemic, 55% of consumers purchased fresh ingredients through comprehensive e-commerce platforms, and more than 76% of the respondents said that they would still purchase ingredients from this channel after the epidemic ends. At the same time, some consumers choose to buy fresh food offline after the epidemic is over. Our study is mainly concerned with the influence of COVID-19 on consumer channel preference. There are some key questions that need to be addressed. How has COVID-19 shaped people’s preference for raw food purchase channels, what are the factors affecting consumers’ online and offline access to food, and what are some of the characteristics of farmers’ markets and online raw food e-commerce that influence consumers’ purchase choices?

Resident behavioral habits play a key role in explaining their purchasing behaviors in purchasing channels. The emergence of COVID-19 has changed people’s lifestyles. High infection and mortality rates have led to stressors, such as fear and anxiety. Many people are worried about contracting COVID-19 and therefore fear contact with people who may be infected with the new coronavirus [6]. Previous studies have found that risk perception is an important factor affecting consumers’ purchase intentions [7]. However, there are few studies focusing on the effect of risk perception of the epidemic on consumers’ choice of fresh food shopping channels. When consumers buy fresh food, quality plays a vital role in their purchase decisions [8]. Many studies have focused on the impact of fresh food quality on consumers’ purchasing behavior [9], but because consumers cannot see fresh products in an online environment, their perceived quality becomes the decisive factor in their purchase. Therefore, unlike existing research, this article explores the impact of perceived quality on purchases and discusses how consumers can obtain information about their perceived quality. From a management point of view, understanding the source of consumers’ perceived quality can help fresh food e-commerce merchants more accurately deliver quality information to consumers and reduce the problem of information asymmetry.

Many studies have demonstrated the impact of consumers’ perceived value on their purchasing behaviors [10]. Fresh food e-commerce is a new way to buy fresh food. Many studies have focused on the impact of online product performance on consumer purchase channels. However, there are few studies that compare different sources of perceived value to compare consumer emphasis on them. Therefore, our research included consumer attention to the five dimensions of quality, food safety, price, logistics efficiency, and platform reputation as independent variables, and observed which of these dimensions receive more attention from consumers and whether these preferences will continue or increase even after the epidemic. In this way, it was possible to identify consumer areas of concern. Our research also produced suggestions for fresh food e-commerce merchants to address their shortcomings and improve their services.

The raging epidemic has raised new questions for the study of fresh food purchase channels. Many studies in the past have found that the price of fresh food and logistics efficiency are important factors that affect consumers’ choice of fresh food purchase channels [11,12]. The COVID-19 epidemic since 2020 has had a profound and serious impact on the global economy. Maintaining social distancing has become a key requirement during the epidemic. After a series of reports on the close connection between offline markets and the epidemic, food safety and delivery safety have become the focus of consumers’ attention [13], and the use of fresh food e-commerce platforms has become an important way to buy fresh food during the epidemic. Many previous studies have shown that online shopping is mostly preferred by young consumers [14].

In China, 93% of netizens born from 1980 to 1995 were found to use online shopping, while 41.9% “Post 95” young online shopping users accounted for over 30% of total daily consumption, and the proportion of consumption of e-shopping was greater than that of the other age groups [15]. Baubonienė et al. [16] (2015) believed that respondents in the twenty-five to thirty-five age group often chose to shop online. Barska et al. [17] (2020) argue that online food purchasers tend to be young and well-educated. Based on research on influential factors of online shopping for older adults, one of the key factors in promoting online shopping for older people is social influence [18]. So, can the epidemic encourage the elderly to cross the digital divide and learn more about mobile phones and the internet, thereby increasing the frequency of their online purchases? What are the factors that influence consumers’ choice of fresh food purchase channels? What should governments and businesses do to meet the needs of consumers?

In this study, we conduct empirical research and analysis to address these questions. This paper includes five sections. Section 2 examines the theoretical background of consumer choice of fresh food purchase channels and proposes hypotheses. Section 3 explains our estimation model and the samples and variables used in the model. After introducing the empirical results in Section 4, we summarize the core results and discussions in Section 5 and acknowledge the limitations of our study.

## 2. Hypotheses

In the past few decades, the popularization of internet technology and the development of mobile technology have brought about the vigorous development of online shopping. Many studies have been devoted to studying consumers’ buying behavior. Bhaiswar et al. [19] (2021) and other studies propose that differences in channel risk perception, price search intentions, and waiting time have a significant impact on consumers’ shift from offline shopping to online shopping. He et al. [20] (2019) suggests that factors such as price, product quality, and service quality will affect consumers’ choices of online and offline purchase channels. When consumers buy fresh food, in addition to these factors, the quality and freshness of fresh food from different channels are important factors considered by consumers when determining their purchasing choices, since fresh food differs from clothing due to its perishable and high consumption-frequency characteristics. The hygiene and quality performance of offline markets will also affect consumers’ choice of fresh produce purchasing channels. Once the epidemic eases, contact will no longer be a restrictive factor for consumers to choose fresh food channels. Similarly, the fresh food performance of online purchase channels will affect consumers’ choice of fresh food purchase channels. Due to the need for self-isolation brought about by the epidemic, consumers’ emphasis on the epidemic will also affect the choice of fresh food purchase channels. Additionally, consumers’ own characteristics also determine their choice of fresh food purchase channels. Therefore, we were mainly concerned with the influence of three factors, namely residents’ behavioral habits, offline market performance, and online product performance, which are the main dimensions of our model.

### 2.1. Consumer’s Preferred Information on Fresh Foods

The emergence of COVID-19 has changed the daily lives of people around the world. The extremely high infection and death rates have led to mental stressors, such as fear, anxiety, and depression. Most people worry about contracting COVID-19, which increases their fear of coming into contact with people who may be infected with the new coronavirus [6]. In the past, for the discussion of consumer behavior references, expectation confirmation theory has been widely used to evaluate consumer satisfaction, post-purchase behavior (such as repurchase or complaint), and general service marketing [21]. Many scholars have verified consumers’ willingness to repurchase products and related continued services using the expectation confirmation theory in many fields, such as repurchase of cars and repurchase of video cameras [22]. In the expectation confirmation theory (ECT), the relationship between expectation and confirmation is negative, which means that when consumers’ expectations are too high and actual performance does not exceed expectations, the degree of confirmation will be lower and indirectly affect consumer satisfaction. Conversely, lower expectations and higher actual performance will increase the degree of confirmation and indirectly increase satisfaction. Consumers will form a pre-purchase expectation about the performance of the product or service they want to buy, and this expectation will affect consumers’ attitude towards the product and their purchase tendency. After the purchase, consumers will have an awareness of the performance of the product based on the actual user experience. When the product performance exceeds expectations, a positive disconfirmation is generated; when the product performance is equal to the expectation, a confirmation is generated; and if the expectation exceeds the performance, a negative disconfirmation is generated. Next, the consumer’s pre-purchase expectations and post-purchase confirmation or non-confirmation affect consumer satisfaction. Finally, the degree of consumer satisfaction affects consumers’ willingness to use it again. The higher the consumer satisfaction, the higher the willingness to continue using it. Whether consumers are willing to repurchase products or continue to use services is a key success factor for the manufacturer of the product or service, and the main factor that affects consumers’ willingness to continue using the product or service is the level of customer satisfaction.

When consumers buy fresh food, quality plays a vital role in their purchasing decisions [8]. Therefore, if fresh products sold online give consumers a better perception of quality, consumers are more likely to choose to buy fresh products online. This article proposes that consumers’ perception of the quality of online fresh food mainly comes from the following three aspects: pictures on product detail pages, online reviews of products, and consumers’ past online purchase experiences. When shopping online, photos are often better than the actual goods received. Therefore, when consumers pay more attention to pictures when buying fresh food, their perceived quality will be higher. According to ECT, the high perceived quality and the actual product quality reduce consumers’ willingness to buy again. Online reviews have always been regarded as an important factor affecting consumers’ online purchase intentions [23]. The perceived quality of online reviews will vary from product to product because consumers will generally consider negative reviews to be more credible than positive reviews [24]. Therefore, we believe that if consumers perceive quality from online reviews, the perceived quality will be low. According to the ECT, the quality of the received goods is better than the perceived quality based on online reviews. Consumers’ willingness to buy the product again will increase. Consumer behavioral intentions for online purchases vary based on their past customer experiences. Positive consumption-related emotions may lead to strong repurchase intentions [25]. A good online purchase experience of fresh food will increase consumers’ perceptions of the quality of online fresh food, which in turn will affect consumers’ willingness to buy the product again. The epidemic has encouraged consumers’ attempts to buy fresh food online, and only a good purchase experience can convince customers to continue using online channels for fresh food purchases. Therefore, we proposed Hypothesis 1 as follows.

**H1:** 
*Consumers, who less trust online pictures and are used to estimating the quality of fresh products by online reviews and purchase experience, tend to increase their frequency of purchasing fresh products in online channels.*


### 2.2. Experience with Offline Channel

If two commodities can replace each other to satisfy the same desire or need, they can be called substitutes for each other. Therefore, we believe that offline markets and online fresh food sales channels are alternative channels for consumers who need to buy fresh food. “2021 China Online Fresh Food Consumption Research Report” from i Reseaich shows that the new crown epidemic has accelerated online purchases of fresh food, but we believe that whether consumers will continue to choose online fresh food purchasing during the epidemic will be affected by the impact of alternative channels. When consumers choose products, they tend to choose channels that offer a better shopping experience [26]. Consumers’ offline purchase experience and market environment conditions can also affect consumers’ offline consumption behaviors [27]. Therefore, we believe that the better performance of offline markets decreases consumers’ attention to purchasing online channels. Thus, we proposed Hypothesis 2a.

**H2a:** 
*The improvement of circumstance conditions in offline markets during COVID-19 decreases consumers’ purchase frequency of fresh products in online channels.*


Meanwhile, food safety is an important criterion for consumers when choosing food [28,29]. Reports on food safety affect consumers’ risk perceptions, and risk perception affects consumers’ purchasing behaviors [30]. The impact of food safety issues on consumers’ purchases of fresh food is already known. Some consumers believe that offline fresh foods are more safe due to the guarantee of the government’s physical inspections [31]. Since the government improved the management of offline markets during COVID-19, consumers who believe that offline fresh foods are safer than online fresh foods tend to rely more on offline channels. Thus, we proposed Hypothesis 2b.

**H2b:** 
*Consumers who believe that the fresh foods in offline channel are much safe than that in online channel tends to decrease purchase frequency of fresh products in online channel.*


### 2.3. Experience with Online Channel

The impact of customers’ perceived value on their behavior is supported by theoretical and empirical research. Research by Sweeney and Soutar [32] (2001) and Molinillo et al. [10] (2021) proved that customer perceived value directly affects their purchase intention, and purchase intention has been proved to be a significant predictor of consumers’ online shopping behavior [33]. Therefore, we believe that consumers’ perceived value of online fresh food will affect the increase in the frequency of their online fresh food purchases during the epidemic. For the purposes of this study, we primarily measured consumers’ perceived value of online fresh foods in terms of their attention to the five dimensions of quality, food safety, price, delivery time, and platform reputation.

Product quality is one of the important attributes of fresh produce and is the core and basic standard of consumer choice. As necessities of life, consumers’ purchase behavior is characterized by high-frequency rigid demand [34]. For food, internal qualities include appearance, color, shape, and structure. External quality constitutes that which is not the physical composition of the product, including price, brand, store name, origin, nutrition, production information, etc. [35]. If the fresh agricultural products sold online make consumers feel better about internal and external quality, consumers are more likely to choose to buy fresh agricultural products online. Many studies have proved that a higher perception of product quality is positively correlated with purchase intention [36]. Cao et al. [37] (2021) suggest that perceived quality has a positive impact on behavioral intentions. Cang et al. [38] (2021) found that the higher consumers’ expectations for product safety and quality, the lower their willingness to buy fresh produce over the internet. When consumers choose to shop, they can compare existing options by weighing costs and benefits [39]. Consumer overall satisfaction will be influenced by perceived product quality or value [40]. In addition, consumers can feel the quality of fresh food in offline markets directly before they buy. When consumers buy fresh food online, they can only indirectly experience the quality of fresh food online due to information asymmetry. As a result, we believed that consumers who are more focused on the quality of fresh food will be more inclined to opt for offline markets. Thus, we proposed Hypothesis 3a.

**H3a:** 
*Consumers’ attention to the quality of online fresh food negatively affects their frequency of buying fresh food online.*


In this study, “food safety awareness” mainly refers to the degree of consumers’ attention to food safety requirements. Many previous studies have concluded that food safety issues are an important factor affecting consumers’ willingness to buy food [41]. Consumers are willing to buy safer foods and pay extra for safer products and food safety labels [42,43]. Zhang et al. [44] (2018) survey found that 67.6% of consumers were willing to buy safe vegetables, 65.8% were willing to pay high prices for safe vegetables, and consumers’ willingness to buy safe vegetables was positively influenced by safety awareness. Zhao et al. [45] (2017) also pointed out that consumers’ attention to food safety will positively regulate the influence of the reference effect on their willingness to buy fresh food online. Because offline markets already have a regulatory system in place, food safety is relatively stable under multiparty monitoring, and consumers have greater confidence in the safety of their fresh foods. The new model of e-commerce for fresh food differs from the offline marketplace. Food product monitoring is not comprehensive, and there are numerous reports of food safety issues in e-commerce of fresh foods. In terms of consumers’ perceptions, the risks to food safety are higher than in offline markets. Therefore, we believed that consumers’ food safety awareness of online fresh food will affect their purchasing behavior. Thus, we proposed Hypothesis 3b.

**H3b:** 
*Consumers’ food safety consciousness (FSC) of online fresh food affects their frequency of online fresh food purchases.*


Price is defined from the consumers’ point of view as “something that is given up or sacrificed in order to get a product” [46]. Price is one of the significant factors affecting consumers’ decisions to purchase food. This can affect consumers’ purchase intent in one of two ways: the higher the price, the greater the monetary sacrifice, the lower consumers’ purchase intentions; or, a higher price results in higher perceived quality and safety, and therefore higher purchase intention [47,48]. Price perception is often regarded as a key factor influencing customers’ purchase intention [11]. The higher the price attractiveness of the product, the higher the value perception and the stronger the purchase intention [36]. Perceived price adversely affects behavioral intentions, and perceived value plays a mediating role between perceived price and behavioral intentions. Nguyen et al. [42] (2019) found that a high price will have a negative effect on consumers’ perceived value of the product. Consumer’s perception of perceived prices can be explained by reference to the theory of fairness. According to this theory, “the parties involved in social exchanges compare with each other the ratios of their inputs into the exchange to their outcomes as a result of the exchange. In their research, Huang et al. [48] (2019) found that while food safety incidents have led consumers to increase their awareness of food safety, consumers who are price-sensitive will still reduce the scale of organic and green food safety purchases when they perceive higher prices. Influenced by the cost of the site and other factors, the price of fresh food online is often lower than that of the offline market. Therefore, we believed that price-sensitive consumers will be more inclined to buy fresh food online. Therefore, we proposed Hypothesis 3c.

**H3c:** 
*Consumers’ attention to online fresh food prices positively affects their frequency of buying fresh food online.*


Logistics is a necessary means of transferring goods from suppliers to consumers. Consumers’ willingness to buy online is directly affected by the quality of the logistics service. E-retailers’ quality of product and service delivery (tangible services) leads to an increase in consumer satisfaction and willingness to repurchase [49]. Particularly for fresh agricultural produce, the logistics distribution process should ensure that produce is complete, fresh, and in a timely manner. The demand for logistics services for fresh agricultural products is higher among consumers, which will largely influence consumers’ willingness to choose fresh e-commerce [38]. Using data collected by French consumers from 2015 to 2016, Kaswengi et al. [40] (2019) found that the quality of logistics services is an effective driver of customer satisfaction. Saving time and energy will have a positive impact on purchase intentions because consumers gain practical value from efficient and timely transactions. Consumers usually want to minimize their shopping costs, including money and time, especially in the case of online shopping [50]. Consumers choose online shopping for convenience and to save shopping time and transportation costs [51]. Higher delivery fees or longer delivery times may make them choose not to shop online. Therefore, high logistics efficiency will encourage consumers to choose to buy fresh food online. While logistics efficiency can be regarded as extremely fast for offline markets, it will take time for online platforms to purchase, distribute and receive fresh food products. Therefore, we believed that consumers who are more concerned about logistics efficiency will prefer to choose offline markets to buy fresh food. Therefore, we proposed Hypothesis 3d.

**H3d:** 
*Consumers’ attention to the delivery time of online fresh food adversely affects their frequency of consumption of online fresh food.*


The reputation of online shopping platforms is an important attribute for online consumers when making purchasing decisions [52]. Hong et al. [53] (2019) showed that online word-of-mouth has a significant impact on the purchasing decisions of both experienced and potential consumers, and that the objective product reviews perceived by consumers are more helpful to their purchasing decisions. Zheng et al. [11] (2020) found that consumers’ concerns about the credibility of electronic suppliers prevented them from frequently buying fresh food online. Cang et al. [38] (2021) studied experienced consumers and potential consumers, respectively, and found that the reputation of online shopping platforms has a significant positive impact on experienced consumers as well as potential consumers’ online purchase of fresh farm produce. Luo et al. [54] (2019) led the way in introducing consumer trust as a mediating variable, and established a model for empirical research based on current online reputation as one of the key factors influencing consumers’ buying behavior. Our results show that consumers’ tendency to trust significantly affects their perceived credibility of online shopping products, and ultimately affects the willingness to shop online via mediation. Consumers can not sense and judge the quality of fresh farm produce in person as part of the e-commerce process, and can only understand products through the display of photographs on the website and the introduction of text [55]. Platform merchants will have more power to show consumers more genuine information about the product for platforms with better reputations. At the same time, in an effort to reduce the potential for information asymmetries, online buyers will tend to purchase products on platforms with higher reputations. Therefore, we proposed Hypothesis 3e.

**H3e:** 
*Consumers’ attention to the reputation of online shopping platforms affects their frequency of buying fresh food online.*


## 3. Data and Questionnaire

### 3.1. Data

The data used in this study came from an online questionnaire survey designed and distributed by our research group. The survey lasted from 19 February to 25 February 2021. Since most of the students had no experience in purchasing fresh products from farmers’ markets, we obtained 20,576 valid questionnaires after deleting students from our sample. During the sample processing process, samples with no changes in online purchases before and during the epidemic and samples with missing values were deleted. The final sample included 10,708 sets of observations.

### 3.2. Questionnaire Structure

The questionnaire for this study consists primarily of four parts. We began with demographic variables, including education level, sex, age, and marital status. The second component was the offline shopping experience, which was measured by the offline market environment [56] and the perception of food safety [45]. The following questions were asked: “How do you think the overall environmental hygiene of offline markets is?” on a Likert scale fixed at 5 (1 denotes strongly satisfied, 5 denotes strongly dissatisfied) level; “Is fresh food in offline markets safer?”, which was defined as a dummy variable of “0–1” (0 denotes no, 1 denotes yes). Third, we examined the source of respondents’ perceived value of purchasing fresh food online, including the use of online images [57], online reviews [38], and the shopping experience [58]. Some specific problems include: “I relay on the fresh food image to estimate the quality of the food”, “I relay on the other consumers’ comments of fresh foods to estimate”, “I relay on my past experience of purchasing of fresh foods to estimate the quality of foods”. Three dummy variables of “0–1” were fixed for this part of the questionnaire (0 denotes no, 1 denotes yes). Fourth, consumers’ online shopping experience, including quality [11,40], security [59], price [11], delivery time [40], and reputation [11] were studied. Examples of entries include “The importance of the quality of food when I decide to buy fresh food in the online channel. “, “The importance of food security when I decide to buy fresh food in online channels. “, and so on. The five variables in this section were fixed as a 10-level Likert scale in the questionnaire (1 denotes strongly disagree, 10 denotes strongly agree) to more accurately collect consumer insights. Table A1 shows all the questions in our questionnaire.

### 3.3. Questionnaire Pre-Test

This study conducted a pre-test of the questionnaire prior to the formal distribution of the questionnaire. The content of the pre-test is primarily divided into two phases. In the first step, expert interviews were conducted on the initial questionnaire, and experts were invited to evaluate the questionnaire structure, questionnaire description and item design, and make amendments to the questionnaire according to relevant suggestions. Secondly, 230 questionnaires were distributed and collated. The reliability and validity of the data collected was verified, and the questionnaire was formally distributed after successful completion of the test.

### 3.4. Reliability and Validity Test

#### 3.4.1. Validity Test

The present study used exploratory factor analysis to test questionnaire validity. The first was to test the correlation between the variables, and the second was to determine whether the sample was suitable for factor analysis by KMO sample test and Bartlett sphere test. Based on the test results, the KMO value was 0.739 > 0.5, which indicates that factor analysis is appropriate. The results of Bartlett’s chi-squared test show that the significance of the chi-squared statistics was 0.000 < 0.01, indicating that the Bartlett’s sphere test results are meaningful and that further research and analysis can be conducted.

#### 3.4.2. Reliability Test

In this study, the internal consistency method was used to verify the reliability of the questionnaire with Cronbach’s Alpha coefficient. We analyzed 10 questionnaire items in the study. Overall, the results revealed that the questionnaire had an overall Cronbach’s Alpha coefficient of 0.735 > 0.6. The questionnaire had high reliability and good consistency.

## 4. Empirical Results

### 4.1. Descriptive Statistics

Table 1 shows the descriptive statistical results of all variables. Out of the 10,708 valid respondents, 77.8% indicated that the frequency of online purchases of fresh food had increased during the epidemic, which also confirms the effect of the epidemic on consumers’ online purchase of fresh food. Descriptive statistics show that the overall education level of the respondents was relatively high, and the gender distribution was relatively even. There were slightly more males than females, with males accounting for 51.7% and females accounting for 48.3%. Most of the respondents were married. Married respondents accounted for 84.3% of the total.

### 4.2. Regression Results

The main problem we investigated in this study was the influencing factors of changes in consumer shopping frequency before and after the pandemic. The dependent variable was a “0 to 1” dummy variable. According to our research purposes and data characteristics, we assigned a value of 1 to “after the COVID-19 epidemic, consumers’ online fresh food purchase frequency has increased compared with that before the epidemic” and a value of 0 to “after the COVID-19 epidemic, consumers’ online fresh food purchase frequency has not increased compared with that before the epidemic”. There are only 1 and 0 in the dependent variable selection scheme set, so we chose the binary logic model for analysis. The results of the estimation parameters from the logic analysis using SPSS are shown in Table 2.

First, we tested the influence of demographic variables as control factors. Consumers with a higher education level will increase the frequency of buying fresh food online under the influence of COVID-19 (*β* = 0.21, *p* < 0.001). Compared to women, men were more affected by the epidemic and increased the frequency of online fresh food purchases (*β* = 0.244, *p* < 0.001). Under the influence of the epidemic, older consumers preferred to choose to buy fresh food online instead of buying fresh food at offline markets (*β* = 0.019, *p* < 0.001); compared to married consumers, unmarried consumers showed a higher increase in the frequency of buying fresh food online under the influence of the pandemic (*β* = −0.173, *p* < 0.05). In the case of consumers’ judgment standards on the quality of online fresh products, consumers who pay attention to online pictures are more likely to buy fresh products online (*β* = −0.311, *p* < 0.05). Customers who pay attention to online evaluations are less likely to buy fresh food online under the influence of the epidemic (*β* = −0.136, *p* < 0.05). And compared with consumers who did not buy based on their purchasing experience, consumers who bought based on their purchasing experience were less affected by the epidemic and increased their frequency of online purchases of fresh food. (*β* = −0.248, *p* < 0.001).

Then, we paid attention to the impact of some offline markets. Surprisingly, the overall environmental sanitation and the quality of fresh products from offline markets have no significant impact on online fresh food purchases (*p* > 0.01). Consumers’ awareness of the safety of online fresh products will significantly affect the increase in the probability of online fresh food shopping (*β* = −0.805, *p* < 0.01).

The empirical results of online fresh food purchasing experience show that consumers who pay more attention to fresh food safety have increased their online purchasing frequency during the epidemic (*β* = −0.033, *p* < 0.05). Consumers’ attention to the reputation of the platform negatively affects the increase in purchase frequency (*β* = −0.037, *p* < 0.01). However, consumers’ concerns about the quality, price, and transportation of online fresh food did not significantly affect the changes in the frequency of consumers’ online shopping (*p* > 0.01). Table 2 and Table A2 presents the results of the estimated parameters of the logic analysis, in which the *β* coefficient and standard error of each variable are reported.

## 5. Discussion

Through a consumer questionnaire analysis of fresh food choices in fresh food e-commerce and offline markets, we found that Chinese COVID-19 pandemic. We used the logic regression model to analyze the increase in the frequency of online fresh food shopping before and during the epidemic for consumers with different characteristics.

Attention to the epidemic resulted only in consumers’ attempts to buy fresh food online. These were not the reasons that influenced consumers to continue to choose online fresh food purchasing. Consumers’ choice of online fresh food shopping is more likely to be determined by factors such as product quality and purchase experience [60]. Therefore, fresh food e-commerce merchants, while attracting consumers to try online purchases through marketing and other means, should pay more attention to the quality of the product itself and the consumer’s buying experience to retain consumers.

Consumers who cared about online reviews significantly reduced their online purchases during the epidemic. According to consumer expectation confirmation theory, improving the quality of online fresh products during the pandemic will help improve consumer quality confirmation. However, studies have found that consumers who are more concerned about online reviews significantly reduce the frequency of online purchases during the epidemic. Although this result is contrary to our hypothesis, it also provides other theoretical logic behind consumer buying behavior. According to the psychological deviation theory of consumer psychology, negative things have a greater impact on people. The increased consumer-perceived risk caused by negative online reviews will cause consumers to abandon online purchases of fresh food. This also explains why many e-commerce merchants increase the frequency of positive reviews from consumers through various methods, such as cashbacks. One of the difficulties and pain points of online shopping is the asymmetry of information. The feedback helps to make the information transparent, thereby avoiding the phenomenon of “bad money driving out good money.” Therefore, online shopping information feedback is a good quality-improvement mechanism. According to the theory of trust, the reliability of information release channels is also an important factor affecting the effectiveness of information transmission. Therefore, to build a good business ecosystem, e-commerce platforms must ensure the reliability of the review information released from the process and enhance the competitiveness of e-commerce.

The overall environmental sanitation and quality of offline markets in consumers’ localities have no significant impact on the increase in the frequency of consumers buying fresh food online under the influence of the epidemic. This finding is surprising. In our traditional perception, the offline market and fresh food e-commerce are alternative channels for each other, and consumers’ choices of one of them will be affected by the situation of the other. This research finding may indicate that the consumer groups who choose to buy fresh food at offline markets in China have a low degree of overlap with those who buy fresh food online. In fact, many studies have proven the existence of consumer stratification in society. Leo et al. [61] (2018) found that typical consumption patterns are closely related to certain socioeconomic classes, leading to stratification patterns in social structures. This is also true for consumers who choose to buy fresh food from offline markets and from fresh food e-commerce platforms. Consumers who buy fresh food in offline markets are older and have more free time. They have been used to buy fresh food from offline markets since the early days. Consumers who like to buy fresh food online are younger and are office workers [20]. Because of the need to go to work, it is difficult for them to find time to go to a offline market to buy fresh food in person. Therefore, buying fresh food online is their first choice. As a new type of e-commerce, fresh food e-commerce has a very low market share compared to traditional e-commerce. Therefore, our research suggests that the marketing focus should be on those who are more likely to become loyal users of fresh e-commerce. For consumer groups, the marketing plan of e-commerce platforms should give more consideration to the characteristics of young, unmarried and highly educated white-collar males, and target marketing to achieve marketing goals.

Many studies have proved the impact of product quality and price on consumers’ purchasing behavior. Surprisingly, our research found that consumers’ attention to the quality and price of online fresh food has no significant impact on the increase in the frequency of consumers buying fresh food online, which is determined by the special nature of fresh food. Compared with clothing and other commodities that are also available for purchase online and offline, the quality of online fresh food and the fresh food available in offline markets has gradually converged. The prices of fresh foods online and offline have also tended to be consistent. At this time, food safety has become a breakthrough in fresh food sales, and our research has demonstrated the impact of food safety. E-commerce businesses should attach great importance to food safety and strive to deliver food safety information to consumers. Consumers’ concerns about food safety can be attributed to the frequent occurrence of food safety scandals across the globe in the past few years. In 2013, for example, Fonterra Group, a dairy giant in New Zealand, launched a recall due to the potential contamination of WPC80, a concentrated whey protein, with botulinum; As of 2017, many Brazilian companies were selling expired meat products; Salmonella infections in children commonly occur worldwide. China is also facing a serious food security problem [62]. In 2008, China’s food industry was shocked by the “Sanlu milk powder scandal”; in the past year of 2022, China again exposed the “soil pit pickle” food safety scandal. As a result of this series of events, consumers have become increasingly concerned about food safety. Food safety can also be understood to have become a significant factor influencing consumers’ shopping choices. More important, successive food-related incidents have undermined the credibility of regulators. In addition, the public’s negative views on food safety have caused a great deal of damage to trust in central and local governments. As a result, the government and relevant regulatory agencies should also pay more attention to food safety and introduce more stringent policies to ensure the food safety of fresh foods.

Consumers’ attention to the efficiency of online fresh food logistics has no significant impact on the increase in their frequency of buying fresh food online under the influence of the epidemic. Fresh food e-commerce industry has significantly developed in China over the last decade, since the establishment of YiGuo Fresh foods in 2005. Many early studies have found that the delivery efficiency of fresh food e-commerce will affect consumers’ online fresh food purchase choices [53]. Along with the development of fresh food e-commerce and the gradual improvement of supply chain construction, the efficiency of fresh food e-commerce distribution is improving. In 2015, only Anxianda, a cold-chain logistics company under Yiguo Fresh foods, provided next-day delivery services. In the United States, there are leading e-commerce enterprises such as FreshDirect and AmazonFresh. In Britain, the leading e-commerce enterprise is Ocado. Hema Fresh dish is the leading e-commerce enterprise in China. As the supply chain model and cold-chain logistics technology improve, these leading fresh-food e-commerce companies can already reach delivery efficiency within half an hour of placing an order. Such a receiving efficiency is no less than going to the offline market to buy fresh food. This is also the reason why China’s online fresh food consumers’ attention to logistics efficiency does not significantly affect the increase in the frequency of consumers buying fresh food online.

Meanwhile, the negative impact of consumers’ attention to the reputation of online shopping platforms on the increase in the frequency of consumers buying fresh food online during the epidemic has also been proved. This shows that online shopping platforms for fresh food products have a low reputation among consumers. Fresh food e-commerce is a typical asymmetric information market. Consumers’ information sources for fresh food are largely the signal transmission of fresh food e-commerce merchants, and the reputation system can improve the trust of the two-sided users of the platform. Therefore, the reputation of the platform determines the consumer’s acceptance of signals from fresh food e-commerce merchants. For companies, the importance of reputation is self-evident. Therefore, for fresh food e-commerce companies, it is urgent to improve the reputation of the platform. The company’s social performance and market risks will have an impact on the company’s reputation. Therefore, platform merchants should ensure the platform’s reputation by ensuring the safety and quality of fresh food and the quality of services.

## 6. Conclusions and Future Research Directions

The data presented in this paper are drawn from a questionnaire survey of Chinese consumers to examine how consumers’ choices of fresh food shopping channels will change following the COVID-19 pandemic and factors influencing consumers’ choice of online and offline shopping channels. To the best of our knowledge, this study has made an important contribution as the first paper to determine the impact of COVID-19 on the frequency of consumers’ increased online shopping for fresh foods. The COVID-19 pandemic in 2020, as a result of self-isolation, closure of farmers’ markets, and disruption of the traditional supply chain of fresh food products made it difficult to get out to the traditional fresh food shopping locations to purchase fresh food. Due to consumers’ demand for fresh food, they attempted to order online to meet their demand for fresh food. This can be said to be a “marketing” of fresh food e-commerce, which has greatly promoted the development of fresh food e-commerce. On the other hand, consumers’ response to this “marketing” also serves as a reference point for e-commerce marketers of fresh foods to formulate future marketing strategies.

Meanwhile, we found that the epidemic did lead to more consumers attempting to purchase fresh foods online. For example, what is the pattern of consumers’ purchases of fresh foods in the future? Therefore, we advance several interesting directions for future work, such as how consumers’ willingness to repurchase fresh foods online changes, and what factors will influence consumers’ willingness to repurchase fresh foods. In contrast, our research samples all come from China. Would the findings of this study change for consumers in other countries? How will the cultures of various countries play a regulatory role in this? Future research should be able to further address these questions.

## Figures and Tables

**Table 1 healthcare-10-02581-t001:** Descriptive statistics of variables used in the estimation (sample = 10,708).

Variables	N	Mean (%)	sd	Min	Max
Channel	10,708	0.778 (77.8%)	0.416	0	1
Education	10,708	4.044	1.003	1	5
Gender	10,708	0.517 (51.7%)	0.500	0	1
Age	10,708	3.008	1.031	1	5
Marital	10,708	0.843 (84.3%)	0.364	0	1
Circumstance	10,708	1.428	0.625	1	5
Food Safety	10,708	1.444	0.517	0	1
Picture	10,708	0.0473	0.212	0	1
Comments	10,708	0.466	0.499	0	1
Experience	10,708	0.228	0.419	0	1
Quality	10,708	8.315	2.107	1	10
Safety	10,708	8.353	2.193	1	10
Price	10,708	7.312	2.108	1	10
Delivery time	10,708	7.514	2.077	1	10
Reputation	10,708	8.424	1.883	1	10

**Table 2 healthcare-10-02581-t002:** Empirical results with logic regressions.

	Variable	*β*	Exp (*β*)	Significance
Demographic variables	Education	0.210 (0.010)	1.233	***
	Gender	0.244 (0.051)	1.277	***
	Age	0.019 (0.003)	1.019	***
	Marital	−0.173 (0.081)	0.841	**
Experience with offline channel	Circumstance	−0.018 (0.040)	0.982	
	Food Safety	−0.805 (0.039)	0.447	***
Factors to estimate fresh food quality	Pictures	−0.311 (0.131)	0.732	**
	Comments	−0.136 (0.063)	0.873	**
	Experience	−0.248 (0.074)	0.708	***
Experience with online channel	Quality	−0.004 (0.017)	0.996	
	Safety	−0.033 (0.016)	1.034	**
	Price	0.021 (0.016)	1.021	
	Delivery time	−0.002 (0.017)	0.998	
	Reputation	−0.037 (0.020)	0.964	*
	_cons	−0.821	0.441	***

Standard errors in parentheses. * *p* < 0.1, ** *p* < 0.05, *** *p* < 0.001.

## Data Availability

The data used in our research is from the questionnaire issued by our research group. If you need our questionnaire and data information, please contact: jennica1002@163.com.

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
