# Peer review of "Consumers’ Channel Preference for Fresh Foods and Its Determinants during COVID-19—Evidence from China"

_healthcare, 2022, doi:10.3390/healthcare10122581_

Round 1
Reviewer 1 Report
Please add more indicators to increase the scinetific robustness and readanility for the international readers.
Author Response
The authors appreciate valuable comments from the reviewers with deep gratitude. In the past few months, the authors have revised the manuscript accordingly with your suggestions. Our responses respecting to each of your suggestions and comments are illustrated point-by-point as follows.
Comment 1. Please add more indicators to increase the scinetific robustness and readanility for the international readers.
Response: To make our design more specific and robust, we have added sections 3.2-3.4 to represent questionnaire design, pre-testing, and reliability and validity test. At the same time, we have incorporated additional evidence and examples globally to support our statement in the main context. The improvements are presented in section 5 (lines 509-513, lines 535-541).
Last but not least, the authors would like to express deep gratitude for your suggestions, which are helpful to improve the quality of the manuscript. We hope all of your concerns are satisfactorily responded in revision and we are very glad to respond if any further concerns.
Reviewer 2 Report
Dear Authors Congratulations on choosing the theme that fits the theme of the Special Issue. The topic is interesting, and the sample is relevant despite referring to a vast universe. The article must be revised to strictly follow the journal's rules, namely, with regard to citations. In the Introduction, they should indicate previous studies that suggest that young consumers mostly make online purchases. Instead of naming section 2. Hypotheses, I called it a literature review that would justify each of the hypotheses. It is important to provide more detailed information about the structure of the questionnaire, question referencing, and information about the pilot test. Best regards
Author Response
The authors appreciate valuable comments from the reviewers with deep gratitude. In the past few months, the authors have revised the manuscript accordingly with your suggestions. Our responses respecting to each of your suggestions and comments are illustrated point-by-point as follows.
Comment 1. Dear Authors Congratulations on choosing the theme that fits the theme of the Special Issue. The topic is interesting, and the sample is relevant despite referring to a vast universe. The article must be revised to strictly follow the journal's rules, namely, with regard to citations. In the Introduction, they should indicate previous studies that suggest that young consumers mostly make online purchases.
Response: In accordance with your suggestion, we found that 93% of Chinese Internet users born between 1980 and 1995 used Internet shopping, and 41.9% of ``post-95"Internet users consumed more than 30%% of their daily consumption. Many studies have also confirmed that young consumers are the dominant force in online shopping. For example, Baubonien et al (2015) argue that consumers in the 25-35 age group tend to shop online, so we think the willingness of young people to shop online is a major trend in the digital age. Based on your suggestion, we incorporated relevant literature (Baubonien et al, 2015; Barska et al., 2020; Lian et al., 2014) and evidence into the fifth paragraph of "1. Introduction" (lines 86-96)
Lian, J. W., & Yen, D. C. (2014). Online shopping drivers and barriers for older adults: Age and gender differences. Computers in human behavior, 37, 133-143.
Baubonienė, Ž., & Gulevičiūtė, G. (2015). E-commerce factors influencing consumers ‘online shopping decision. Social technologies, 5(1), 62-73.
Barska, A., & Wojciechowska-Solis, J. (2020). E-consumers and local food products: A perspective for developing online shopping for local goods in Poland. Sustainability, 12(12), 4958.
Comment 2. Instead of naming section 2. Hypotheses, I called it a literature review that would justify each of the hypotheses. It is important to provide more detailed information about the structure of the questionnaire, question referencing, and information about the pilot test.
Response: To clarify the research hypothesis, we divide hypothesis 3 into five hypotheses: H3a, H3b, H3c, H3d, and H3e. At the same time, in accordance with your suggestion, we looked at more previous literature and found that many articles will introduce more about the design of the questionnaire. Zhang et al. (2019) detailed the questionnaire design and reference. Chaudhary et al. (2018) detailed their pre-test and reliability and validity test questionnaire. Refer to the existing studies, we have specify ou the questionnaire structure, questionnaire design, pre-test, reliability and validity test in more detail. Our improvements are described in sections 3.2-3.4.
Cheung, M. F., & To, W. M. (2019). An extended model of value-attitude-behavior to explain Chinese consumers’ green purchase behavior. Journal of Retailing and Consumer Services, 50, 145-153.
Chaudhary, R., & Bisai, S. (2018). Factors influencing green purchase behavior of millennials in India. Management of Environmental Quality: An International Journal.
Last but not least, the authors would like to express deep gratitude for your suggestions, which are helpful to improve the quality of the manuscript. We hope all of your concerns are satisfactorily responded in revision and we are very glad to respond if any further concerns.
Reviewer 3 Report
There are many typos and grammar errors. I also do not know what "Agricultural food" is in this context. Please define. What is non-agriultural food?
The major problem is with the regression. First, just present the final model (Model 4?)
More seriously, ordinal variables do not make good regressors because they have no units. A one point change in a scale for one respondent is not the same as for another. Better to create binary dummy variables (low-high) and use those.
Code education as number of years. (like 8 for Jr High school, 12 for high school, 12 for college degree, etc.). As it is now, a graduate degree (5) is five times more education than Junior High (1), which is misleading and makes the results meaningless. For age, code as the midpoint of the category (like "22" for 18-26, "30" for 26-35, etc. When you code 1 for age 25 and 5 for age 57, it means 57 is five times greater than 25 which is again false and misleading.
In each case you need to make a binary or a true ratio variable with a unit (years of education) for the results to be accurate
Author Response
The authors appreciate valuable comments from the reviewers with deep gratitude. In the past few months, the authors have revised the manuscript accordingly with your suggestions. Our responses respecting to each of your suggestions and comments are illustrated point-by-point as follows.
Comment 1.There are many typos and grammar errors. I also do not know what "Agricultural food" is in this context. Please define. What is non-agriultural food?
Respond: The research subjective products are the outputs from farms, such as fresh vegetables, fruits, potatoes and beef. We searched many related literatures and found 'fresh food' represents the products that we focus on. For example, Nakandala et al. (2016) aimed to reduce the cost and improve quality of "fresh food" based on games between farms and retailers. Kaipia et al. (2013) studied information sharing in the fresh food supply chain. Referring the existing articles and replace "agricultural foods" by "fresh food".
Nakandala, D., Lau, H., & Zhang, J. (2016). Cost-optimization modelling for fresh food quality and transportation. Industrial Management & Data Systems.
Kaipia, R., Dukovska‐Popovska, I., & Loikkanen, L. (2013). Creating sustainable fresh food supply chains through waste reduction. International journal of physical distribution & logistics management.
In order to improve the language, we employ a company named AJE to edit our languages. Hope all typos and grammar errors have been corrected.
Comment 2. The major problem is with the regression. First, just present the final model (Model 4?)
Respond: According to your valuable suggestions, we just present final model 4 in the revised manuscript which helps to illustrate the results more clearly and concisely. The revised table is listed on page 9-10.
Comment 3. More seriously, ordinal variables do not make good regressors because they have no units. A one point change in a scale for one respondent is not the same as for another. Better to create binary dummy variables (low-high) and use those. Code education as number of years. (like 8 for Jr High school, 12 for high school, 12 for college degree, etc.). As it is now, a graduate degree (5) is five times more education than Junior High (1), which is misleading and makes the results meaningless. For age, code as the midpoint of the category (like "22" for 18-26, "30" for 26-35, etc. When you code 1 for age 25 and 5 for age 57, it means 57 is five times greater than 25 which is again false and misleading. In each case you need to make a binary or a true ratio variable with a unit (years of education) for the results to be accurate
Respond: Thank you very much for your correction and guidance. In the questionnaire, age is asked by sector and education was asked at the highest level. Based on your suggestion and some literature (Steenkamp et al.., 2002), we change the age to median and replace education with the number of years of schooling. So, the age range between 18-25 years old is given a value of 20, 26-35 years old is 30, 36-45 years old is considered to be 40, 46-55 years old is given a value of 50 and 56 years old and above is considered to be 60. The value of "education" is assigned as: 9 for junior high school and below, 12 for high school (including vocational high school), 14 for college education (including vocational colleges), 16 for undergraduate degree and 19 for graduate degree and above. We show the corrected regression results in Table 2.
Steenkamp, J. B. E., & Burgess, S. M. (2002). Optimum stimulation level and exploratory consumer behavior in an emerging consumer market. International Journal of Research in Marketing, 19(2), 131-150.
Last but not least, the authors would like to express deep gratitude for your suggestions, which are helpful to improve the quality of the manuscript. We hope all of your concerns are satisfactorily responded in revision and we are very glad to respond if any further concerns.
Reviewer 4 Report
The article presents an important issue of consumers’ channel preference for agricultural food products in China during COVID-19. The paper is well-elaborated and well-structured. The literature review is robust, and the reasoning is adequately conducted.
However, the description of the applied methods, especially the statistical tools, is not sufficiently detailed, and the justification for their use is missing. Also, the results and their implications could be wider described. The Introduction section lacks a clearly stated objective of the study, and the Conclusions – recommendations for further research.
Author Response
The authors appreciate valuable comments from the reviewers with deep gratitude. In the past few months, the authors have revised the manuscript accordingly with your suggestions. Our responses respecting to each of your suggestions and comments are illustrated point-by-point as follows.
Comment 1. The article presents an important issue of consumers’ channel preference for agricultural food products in China during COVID-19. The paper is well-elaborated and well-structured. The literature review is robust, and the reasoning is adequately conducted. However, the description of the applied methods, especially the statistical tools, is not sufficiently detailed, and the justification for their use is missing.
Respond: Thank you very much for your recognition of our article. Our dependent variable is: "Whether the online agricultural food purchase frequency of consumers after the epidemic has increased compared with that before the epidemic". "Increase" is assigned a value of 1, otherwise it is assigned a value of 0. Since our dependent variable is a "0-1" dummy variable, it meets the use conditions of the binary logistic regression model. Therefore, we chose binary logistic regression model for our research. We used SPSS software for regression and obtained our regression results (table 2). According to your valuable opinions, we added our statistical tools and the reasons for choosing binary logistic regression model in the first paragraph of 4.2 (lines 399-408).
Comment 2.Also, the results and their implications could be wider described.
Respond: Thank you for your suggestions. In the discussion section of Chapter 5, we discussed in detail the significance of each conclusion of our research, including the practical significance of the assumptions supported in the research. For example, the negative impact of consumers' concern about the reputation of online shopping platforms on the increase in the frequency of online shopping for fresh food after the epidemic has also been proved. Therefore, for fresh food e-commerce, it is extremely urgent to improve the reputation of the platform. And, we also explain the unsupported assumptions. In order to make our "5. Discussion" part more acceptable, we have added some international examples, such as lines 509-513, lines 535-541, etc. I hope it can make our discussion more acceptable.
Comment 3. The Introduction section lacks a clearly stated objective of the study
Respond: Our study aims to explore whether public channel preferences for fresh foods have changed and to identify potential determinants. So, the goal of our research is to answer "Can the empirical environment the elderly to cross the digital divide and learn more about mobile phones and the internet, there increasing the frequency of their online purchases? What are the factors that influence consumers' choice of fresh food purchase channels?" And so on. In response to your suggestion, we discussed further the objectives of the second paragraph (lines 46-49) and fifth paragraph (lines 96-99) of "1. Introduction".
Comment 4. the Conclusions – recommendations for further research.
A: This study explored the changes in consumers' choice of fresh purchase channels and their influencing factors after the pandemic. Thus, we put forward several interesting further research directions in the future. For example, are consumers willing to buy fresh food online? What factors will affect consumers' willingness to buy back fresh food? Will the findings change for consumers in other countries? What regulatory role will different cultures play in this? At your suggestion, we added "6." Conventions and future research directions" after "5. Discussion". (lines 560-580)
Last but not least, the authors would like to express deep gratitude for your suggestions, which are helpful to improve the quality of the manuscript. We hope all of your concerns are satisfactorily responded in revision and we are very glad to respond if any further concerns.
Round 2
Reviewer 3 Report
There are still some typographical errors (like "Logic" not "Logit" model)
major concerns addressed
Author Response
There are still some typographical errors (like "Logic" not "Logit" model)
Respond: Thank you so much for the suggestion. Guided by your suggestions, we revised the name of Logit to Logic in our statements. We have also reviewed the entire article and corrected the typos found. Many thanks for your suggestions, this helps a lot.
major concerns addressed
Respond: Thank you very much for the suggestion. Online shopping has provided another purchase channel outside the traditional shopping environment and COVID-19 pandemic has created new opportunities for fresh food e-commerce. Our study is mainly concerned with the influence of COVID-19 to consumer’s channel preference. Therefore, we try to answer three questions in specific. How has COVID-19 shaped people's preference for raw food purchase channels, what are the factors affecting consumers' online and offline access to food, and what are some of the characteristics of farmers' markets and online raw food e-commerce that influence consumers' purchase choices? Following your suggestion, we present our main concerns and research questions in introduction section (Lines 46-51).